# Intrinsic negative magnetoresistance from the chiral anomaly of multifold fermions

Federico Balduini [1,4] ✉, Alan Molinari[1,4], Lorenzo Rocchino[1], Vicky Hasse[2], Claudia Felser [2], Marilyne Sousa[1], Cezar Zota[1], Heinz Schmid [1], Adolfo G. Grushin [3] ✉ & Bernd Gotsmann [1] ✉

The chiral anomaly - a hallmark of chiral spin-1/2 Weyl fermions - is an imbalance between left- and right-moving particles that underpins phenomena such as particle decay and negative longitudinal magnetoresistance in Weyl semimetals. The discovery that chiral crystals can host higher-spin generalizations of Weyl quasiparticles without high-energy counterparts, known as multifold fermions, raises the fundamental question of whether the chiral anomaly is a more general phenomenon. Answering this question requires materials with chiral quasiparticles within a sizable energy window around the Fermi level that are unaffected by extrinsic effects such as current jetting. Here, we report the chiral anomaly of multifold fermions in CoSi, which features multifold bands within ~0.85 eV of the Fermi level. By excluding current jetting through the squeezing test, we measure an intrinsic, longitudinal negative magnetoresistance. We develop a semiclassical theory to show that the negative magnetoresistance originates in the chiral anomaly, despite a sizable and detrimental orbital magnetic moment contribution. A concomitant non-linear Hall effect supports the multifold-fermion origin of the magnetotransport. Our work confirms the chiral anomaly of higher-spin generalizations of Weyl fermions, currently inaccessible outside solid-state platforms.

The spin-statistics theorem forces elemental fermions to have half-integer spin. For instance, the Hamiltonian of a three-dimensional spin-1/2 massless Weyl fermion is $H = \eta \hbar v \mathbf{k} \cdot \boldsymbol{\sigma}$, which is linear in momentum $\mathbf{k}$, has a characteristic velocity $v$, and is written in terms of Pauli matrices $\boldsymbol{\sigma}$ that encode its spin-1/2 degree of freedom. The parameter $\eta = \pm 1$ defines the chirality of the Weyl fermion, which acts as sources and sinks, i.e., monopoles, of Berry curvature, with a sign determined by $\eta$[1].

Intriguingly, some structurally chiral materials can realize, as low energy quasiparticles, higher-spin generalizations of Weyl fermions[2–5]. Referred to as multifold fermions, they are governed by a Hamiltonian $H = \eta \hbar v \mathbf{k} \cdot \mathbf{S}$ where $\mathbf{S}$ represents an effective spin degree of freedom. Because $\mathbf{S}$ can represent matrices of any spin, including integer spin,

these massless fermions cannot exist as elementary particles. Nonetheless, multifold fermions also have a definite chirality and act as sources or sinks of Berry curvature. For example, the central, middle and top bands of a spin-1 fermion have associated monopole charges $C = -2, 0, 2$ contrary to Weyl fermions, whose bands are Berry monopoles of charge $C = \pm 1$.

Chiral massless particles are distinguished from non-chiral particles by their response to external magnetic fields. Applying an electric $\mathbf{E}$ and magnetic field $\mathbf{B}$ such that $\mathbf{E} \cdot \mathbf{B} \neq 0$, changes the relative density of positive and negative chirality quasiparticles. This results in a finite chiral current, not expected on classical grounds where positive and negative chirality quasiparticle densities are equal[6]. This quantum effect is referred to as the chiral anomaly. The chiral anomaly of Weyl

[1]IBM Research Europe - Zurich, Säumerstrasse, Ruschlikon, Switzerland. [2]Max Planck Institute for Chemical Physics of Solids, Nöthnitzer Strasse 40, Dresden, Germany. [3]Univ. Grenoble Alpes, CNRS, Grenoble INP, Institut Néel, 25 Av. des Martyrs, Grenoble, France. [4]These authors contributed equally: Federico Balduini, Alan Molinari. ✉e-mail: ico@zurich.ibm.com; adolfo.grushin@neel.cnrs.fr; bgo@zurich.ibm.com

fermions is by now text-book material, first discovered as a contribution to pion decay[6].

In contrast, because multifold fermions do not exist as elementary particles, their anomalies are much less studied theoretically[7–9] and have no experimental confirmation. Experimentally, one challenge lies in identifying a chiral semimetal that realizes chiral massless quasiparticles within a large window of energy around the Fermi energy. Even more dramatically, the main experimental consequence of the chiral anomaly, a longitudinal negative magnetoresistance[10], is often masked by a trivial effect known as current jetting[11,12], arising from an enhanced anisotropic conductivity in a magnetic field in metals.

Nevertheless, the chiral anomaly is theoretically expected for multifold fermions[7,8]. In the ultraquantum limit of high-magnetic fields[3,7], a number of chiral Landau levels equal to the monopole charge of the band can pump left-moving to right-moving chiral fermions, just as in the case of Weyl fermions. However, the ultraquantum limit remains inaccessible in real multifold materials. For small magnetic fields, a semiclassical transport theory, along the lines of derivations for Weyl fermions, predicts a negative magnetoresistance rooted in the Berry curvature[9] (see Figs. 1b, c). However, existing semiclassical derivations neglect the orbital magnetic moment of multifold fermions[13], the self-rotating motion of a wave packet around the magnetic field[14], whose effect can be as sizable as the Berry curvature[15]. It is thus unknown if their combined effect allows negative magnetoresistance arising from the chiral anomaly in multifold fermions to be observable.

Here we report the observation of the chiral anomaly in multifold fermions by measuring magnetotransport in cobalt monosilicide (CoSi) and comparing it to a semiclassical theory that includes the Berry curvature and orbital magnetic moment. We find that the orbital moment decreases but does not overcome the chiral anomaly contribution. In agreement with our theoretical calculations, our measurements reveal a positive longitudinal magnetoconductance up to 1% between [−2.5, 2.5] Tesla. Its angular dependence is well described by the expected but rarely observed $\cos^2\theta$ dependence, where $\theta$ is the relative angle between electric and magnetic fields. These effects occur concomitantly with a non-linear contribution to the Hall effect, originating in both the Berry curvature and orbital magnetic moment, further supporting the multifold-fermion origin of the positive magneto-conductance. We confirm that these properties are intrinsic to the material, as we can discard current-jetting effects using a recently proposed squeezing test[16].

## Results

CoSi is an example of a chiral crystal with multifold fermions at the Fermi level, Fig. 1. It belongs to a family of chiral crystals whose crystal symmetries enforce multifold fermions as low energy quasiparticles. The cleanest experimentally confirmed multifold materials are the chiral crystals AlPt and the monosilicides CoSi, RhSi, all in space-group 198[17–23]. Among them, CoSi is remarkably simple. Ignoring spin-orbit coupling, which is relatively weak compared to RhSi and AlPt, CoSi has been confirmed to display multifold fermions within a large window of

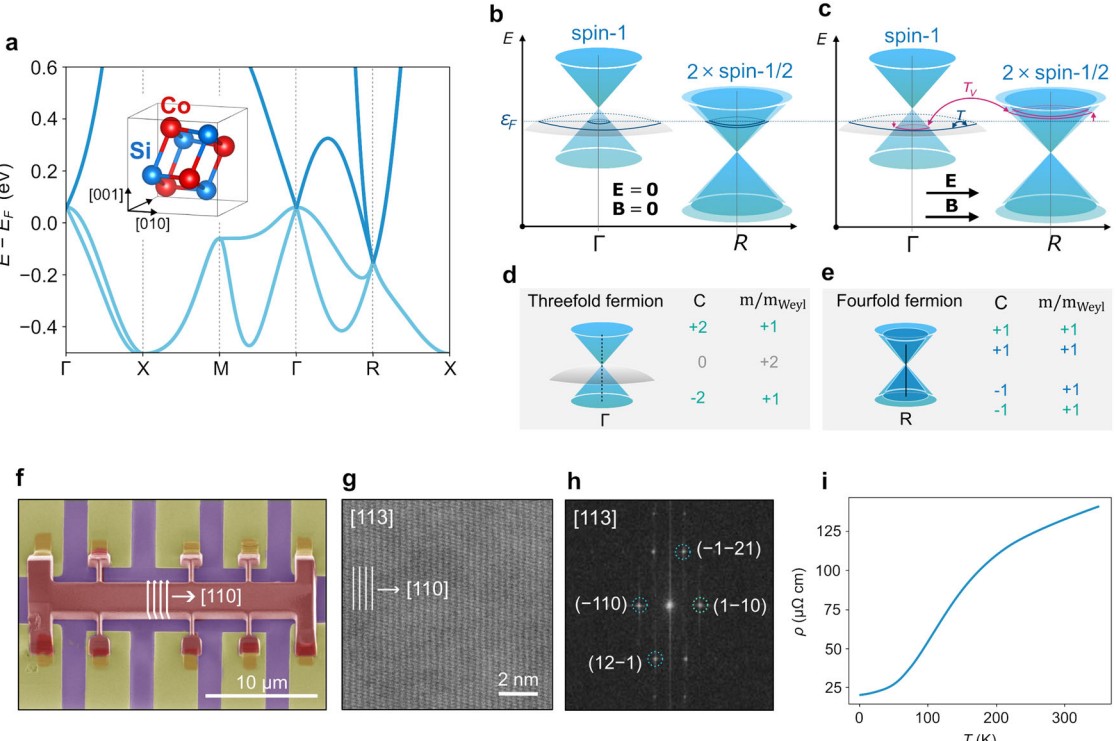

**Fig. 1 | Multifold fermions in CoSi and single-crystal sample. a** Tight-binding band structure of CoSi crystal (inset) without spin-orbit coupling. The spin-1, threefold fermion and the double-spin-1/2 Weyl are located around Γ and R, respectively. **b** The linear and quadratic bands of the spin-1 threefold fermion at Γ and the double spin-1/2 Weyl around R at equilibrium. **c** The chiral anomaly unbalances the density of spin-1 and double-spin-1/2 multifold fermions when $\mathbf{E}\cdot\mathbf{B}\neq 0$ through an internode scattering time $\tau_\nu$, resulting in a negative magnetoresistance. Internode scattering (with characteristic time $\tau$) and a finite orbital magnetic moment contribute against the chiral-anomaly-induced negative magnetoresistance. **d** The linear bands of the threefold fermion at Γ have monopole charge $C=\pm 2$ and orbital magnetization equal to a Weyl fermion ($m/m_{Weyl}=+1$).

The parabolic band at Γ has no Chern number ($C=0$) and double orbital magnetization compared to a Weyl fermion ($m/m_{Weyl}=+2$). **e** The double spin-1/2 multifold fermion at R is composed of two Weyl fermions of equal chirality, separated for clarity. Each has a monopole charge of $\pm 1$ and orbital magnetization $+1$. **f** False-colored scanning electron micrograph of our sample. The multi-terminal Hall bar was cut from a bulk single crystal using focused ion beam sculpturing. **g** Scanning transmission electron microscopy (STEM) shows the atomically resolved structure of the crystal as seen perpendicular to the transport direction, indicating a [110] transport direction. **h** Two-dimensional Fourier transform of the STEM image. **i** Resistivity versus temperature of the device in **e**.

order 0.85 eV around the Fermi level (see Fig. 1a). The band structure obtained from a symmetry-compatible tight-binding model of this material[13,24], fitted to ab-initio calculations[25,26], shows a three-band spin-1 fermion at the Brillouin center (Fig. 1d), and a four-fold crossing at the Brillouin corner R point composed by two copies of a spin-1/2 Weyl fermion of equal chirality that meet in a single point (Fig. 1e). This band-structure has been confirmed by ARPES measurements[17,19] and explains well features seen in optical experiments[13,18,25,26].

For our transport experiments, we grew CoSi single crystals using the chemical vapor transport method. Using a focused ion beam, we fabricated a micro-Hall bar starting from a lamella extracted from the CoSi single crystal, Fig. 1f. The microfabrication allows for good control of geometry and crystalline direction and an even distribution of the magnetic field over the sample. To ensure accurate alignment of the sample, we extracted a second lamella from the same single crystal and oriented it in the same direction as the one used for the Hall bar fabrication. The second lamella underwent scanning transmission electron microscope (STEM) analysis to verify the alignment of the crystalline axes (Fig. 1g–h).

In Fig. 1i, we show the resistivity as a function of temperature. We observe that at low temperatures, the resistivity is $\rho_{xx}(2K) = 20\ \mu\Omega\text{cm}$, and monotonically increases with temperature ($RRR = \rho(300\,\text{K})/\rho(2\,\text{K}) \approx 6$), which aligns with the literature values from bulk crystals, with similar growth conditions and same transport direction (see for example sample I04 in ref. 27). Typically, the chiral anomaly is observed in Weyl semimetals with semiconducting-like resistivity[28–32], indicating the proximity of the Fermi level to the Weyl points. However, this is not the case for CoSi. Due to its large topologically nontrivial energy window, it becomes feasible to observe multifold-related effects even at relatively high carrier densities. This behavior sets CoSi apart from traditional Weyl semimetal systems.

To further characterize the carrier types and electronic pockets at the Fermi level, we now apply a magnetic field perpendicular to the current direction. We observe the resistivity increases quadratically up to 24% at 9 T and 2 K, with no sign of saturation (Fig. 2a). Shubnikov–de Haas (SdH) oscillations are present at frequencies of 21, 556, and 657 T (Fig. 2b). Following previous studies[27,33–35], we assign these frequencies to the tiny hole pocket at $\Gamma$ and the double-Weyl electron pockets at $R$, respectively, indicating the presence of multifold fermions in our sample. The Onsager relation, in combination with the Luttinger theorem, allows extracting the carrier density in $\Gamma$: $n_\Gamma = 5.4 \cdot 10^{17}\text{cm}^{-3}$ and in $R$: $n_R = 1.7 \cdot 10^{20}\text{cm}^{-3}$, in excellent agreement with the value found from

the Hall effect at low temperatures $n_H = 1.7 \cdot 10^{20}\text{cm}^{-3}$, which reveals an electron dominated transport (Supplementary Fig. 2).

Having characterized our sample, Fig. 2a shows the longitudinal magnetoresistance (MR). When the magnetic field is rotated in the direction parallel to the electrical current, the MR exhibits a negative trend, decreasing as $B^2$ until it saturates at approximately 2.5 T. It is worth noting that previous research reported both positive longitudinal MR in CoSi single crystal[36] (yet asymmetric with the magnetic field), and negative longitudinal MR (in CoSi single crystal in ref. 34 and in Fe-doped CoSi in ref. 37.) It appears for a convincing claim on the observation of the anomaly, more detailed cross-checks and modeling is necessary, which we aim to provide in the following.

The chiral anomaly is expected to leave an imprint as a negative longitudinal magnetoresistance[10]. A negative magnetoresistance has been observed in the Dirac semimetals $Na_3Bi$[30], $Cd_3As_2$[31]; in the type I Weyl semimetal TaAs[38,39], NbAs[32] and NbP[29]; in the type II Weyl semimetal $WTe_2$[40]; in the heavy fermion semimetal YbPtBi[41]; in GdPtBi[28] and $ZrTe_5$[42] where Zeeman energy leads to band crossings and the formation of Weyl nodes. In all of these materials, Weyl nodes are fairly close in momentum space and constrained to a small energy window ($\approx 50 - 100$ meV). In comparison, CoSi is advantageous because multifold fermions of opposite chiralities are maximally separated in the Brillouin zone and exist in a large energy window ($\approx 0.85$ eV).

However, before analyzing the longitudinal magnetoresistance, it is necessary to confirm that it is intrinsic. Notably, a longitudinal magnetic field that enhances the anisotropy of the conductivity tensor is known to lead to a spurious negative or positive magnetoresistance depending on the location of the contacts[11,12,43]. To exclude this phenomenon, known as current jetting, we performed the squeeze test proposed in ref. 12. The squeeze test compares different current inlet and voltage probe geometries of the CoSi microbar. Figure 3a shows the most divergent results. The measurements consistently show a negative longitudinal magnetoresistance, regardless of the chosen contact configuration, providing evidence of negligible current jetting[12]. The absence of current jetting phenomena is also consistent with the relatively modest electron mobility of CoSi extracted from the Hall effect $\mu_{2K} = 3 \cdot 10^3$ cm$^2$/Vs (see Supplementary Fig. 2).

We note that other phenomena, such as localization effects, mobility fluctuations, magnetism, or finite-size effects, could potentially contribute to negative longitudinal magnetoresistance. However, we have ruled out these possibilities due to the smooth parabolic behavior observed in both transverse and longitudinal MR at low temperatures,

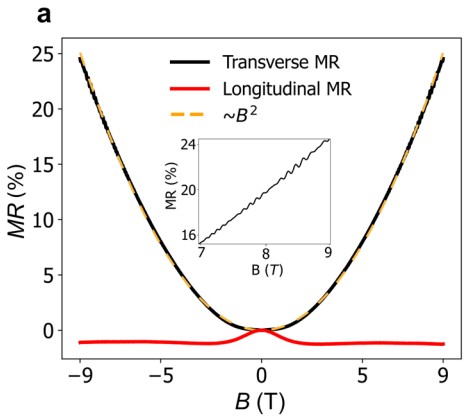
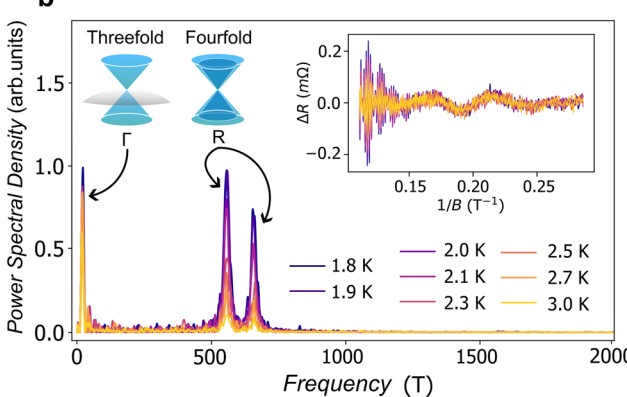

**Fig. 2 | Magnetoresistance of CoSi. a** Magnetoresistance ($MR = (R(B = 0) - R(B))/R(B = 0)$) in transversal geometry ($B \perp I$, black line) and longitudinal geometry ($B \| I$, red line), at 2 K. The approximate parabolic shape of the transversal MR (TMR) is demonstrated using a parabolic fit (dashed yellow line). The inset shows Shubnikov–de Hass oscillations in the TMR. **b** Analysis of the Shubnikov–de Hass oscillations. The residual resistance change $\Delta R$ after subtracting the background

plotted versus the inverse of applied magnetic field $B$ shows several frequency contributions (see inset). The power spectral density of $\Delta R$ in arbitrary units shows three clear frequencies for temperatures between 1.8 and 3 K. These can be assigned to the three and fourfold fermions at $\Gamma$ and $R$, respectively, confirming that both fermions contribute to transport.

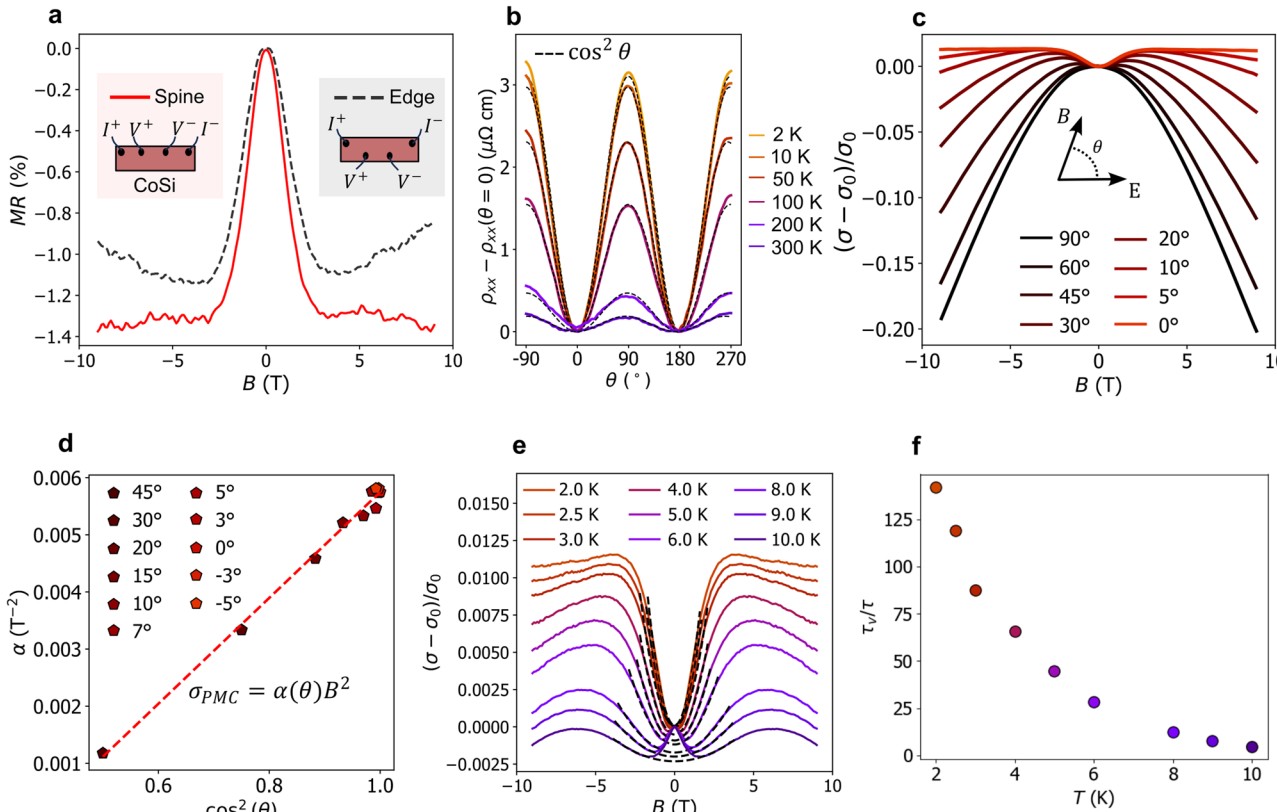

**Fig. 3 | Negative longitudinal magnetoresistance from the chiral anomaly of multifold fermions. a** Result of the squeeze test. The small variation of the longitudinal magnetoresistance upon changing the probing contact geometries demonstrates that the measured phenomenon is intrinsic to the material. **b** Dependence of the resistivity at an applied magnetic field of $B = 9$ T on the angle $\theta = \angle(B, E)$ at various temperatures. Dashed lines are fits to the expected $\cos^2(\theta)$ dependence. **c** Normalized conductance variation measured at 2 K, as a function of

the applied magnetic field for some angles $\theta$ between $B$ and $E$. **d** At small angles $\theta$, the magnetoconductivity increases quadratically with magnetic field $(\sigma - \sigma_0)/\sigma_0 = \alpha B^2$. The parabolic coefficient $\alpha$ increases as $\cos^2(\theta)$, as expected for the chiral anomaly. **e** Longitudinal magnetoconductance at various temperatures together with parabolic fits (black dashed lines). **f** Internode scattering time $\tau_\nu$ over intranode scattering time $\tau$ as function of temperature extracted using the fits in **e** and Eq. (1).

yet with opposite signs, contrary to what is expected in the case of localization effects[44], disorder[45], or mobility fluctuations[46]. The consistently monotonic trend of the negative MR of CoSi at 2 K indicates that finite-size effects do not underlie its origin[43]. Finally, while an excess of cobalt in CoSi ($Co_{1-x}Si_x$) could contribute to the negative MR because of its magnetic properties, such an occurrence would typically coincide with hysteresis in the MR or Hall effect data[47], which is not evident within our experimental error (see Supplementary Fig. 3).

In Fig. 3b we show the change in resistivity as a function of angle $\theta$ between the current and the magnetic field, fixed at 9 T. The experimental data show a clear $\cos^2\theta$ dependence, indicated by the dashed black lines. We observe that, when the magnetic field is aligned with the current, the positive magnetoconductance phenomenon is maximal (Fig. 3c). The positive magnetoconductance increases as $\cos^2\theta$ for $\theta$ approaching 0°, as demonstrated in Fig. 3d.

While the $\cos^2\theta$ is expected for the negative magnetoresistance-induced chiral anomaly, it contrasts with what is observed in Weyl materials, which show a strong narrowing ($\cos^n\theta$ with $n > 2$)[12,30,38], which has been attributed to anisotropy in disorder[48] or magnetotransport[49]. To our knowledge, the only exception is the heavy-fermion semimetal YbPtBi[41]. Narrowing effects seem to be absent or weak in our samples and are consistent with the absence of spurious effects, further supporting that we are measuring an intrinsic contribution.

To interpret the magnetoresistance data we have developed a semiclassical theory for magnetoresistance[10] to the case of multifold fermions. The main difference compared to previous semiclassical

work is the inclusion of the orbital magnetic moment. This may seem a nuanced point, but, as noticed in refs. 15,50 for the case of spin-1/2 Weyl fermions, the orbital magnetic moment contributes with a magnitude comparable to the Berry curvature to the magnetoresistance of Weyl fermions. Hence, in the multifold case, it can cause a sizable under- or over-estimation of the effect or even completely suppress negative magnetoresistance.

Including all contributions, we find that the magnetoresistance is dominated by the spin-1 threefold linear hole band at $\Gamma$ and the double-spin-1/2 electron pocket at $R$, given by (see Supplementary Information)

$$\frac{\sigma - \sigma_0}{\sigma_0} = \frac{(a + b\frac{\tau_\nu}{\tau})\cos^2\theta - c}{20(1 + 2r_2)} \frac{B^2}{(k_F^\Gamma)^4} \frac{e^2}{\hbar^2}, \quad (1)$$

where $a = -9 + 2r_1$, $b = 10(3 + r_1)$, $c = (2 + 4r_1)$, $r_1 = \left(\frac{k_F^\Gamma}{k_F^R}\right)^2 \frac{v_R}{v_\Gamma}$ and $r_2 = \left(\frac{k_F^R}{k_F^\Gamma}\right)^2 \frac{v_R}{v_\Gamma}$. If the ratio between inter ($\tau_\nu$) and intranode ($\tau$) scattering times is $\frac{\tau_\nu}{\tau} > 1$ this implies $\frac{\sigma - \sigma_0}{\sigma} > 0$ for any $B$. At $\theta = 0$ the magnetoconductance increases quadratically with the magnetic field, as seen in our samples, Fig. 3e. The calculated angle-dependent conductivity, proportional to $\cos^2\theta$, is also in agreement with our experiment, Fig. 3d. By tracking the orbital magnetic moment contributions to the conductivity, we observe that these work against, but crucially do not overcome, the positive chiral anomaly terms. Hence, the observation of intrinsic positive magnetoconductance is a signature of the chiral

anomaly of multifold fermions, even when we take into account the large orbital magnetic moment of the multifold fermions.

To be more quantitative, we can make use of the fact that $k_F^\Gamma = 0.25\,\text{nm}^{-1} \approx 5.2\,k_F^R$, which we extract from our quantum oscillation measurements. The tight-binding model fitted to ab-initio calculations shows that $v_\Gamma = v_R/\sqrt{3}$[25,26]. Using these numbers, we can extract the ratio $\tau_v/\tau$ by fitting $(\sigma - \sigma_0)/\sigma_0$ to our measurement. The fit is shown in Fig. 3e, which leads to $\tau_v/\tau \approx 10^2$ at low temperatures (Fig. 3f). The large ratio $\tau_v/\tau$ is indicative of a long-lived chiral current, as expected for multifold fermions with maximally separated chiral branches. For comparison, in their original work on the chiral anomaly in $Na_3Bi$, Xiong et al. found an axial current relaxation time that exceeds the Drude relaxation time by a factor of 40–60[30], yet leading to a larger positive magnetocundactance when compared to CoSi (around 25% in $Na_3Bi$ against the 1% of CoSi). This apparent discrepancy arises because the main contribution to the positive magnetocundactance induced by the chiral anomaly in CoSi comes from the tiny pocket at Γ. The Fermi level is very close to the node at Γ, making the Berry curvature contribution significant compared to that of the R point, which behaves like a large metallic pocket. Indeed, according to Eq. (1) the long

scattering time $\tau_v$, is compensated by the small ratio $r_1 = \left(\dfrac{k_F^\Gamma}{k_F^R}\right)^2 \dfrac{v_R}{v_\Gamma}$, leading overall to a small positive magnetoconductance.

To further validate the multifold origin of the observed negative longitudinal magnetoresistance, we conducted angle-dependent Hall effect measurements to search for indications of a topological Hall effect, which would be indicative of non-zero Berry curvature and non-trivial material topology, given the non-magnetic nature of CoSi. After subtracting a linear fit from the Hall data shown in Fig. 4a, we observe a cubic-in-$B$ contribution to the Hall effect, as depicted in Fig. 4b–e. Remarkably, the observed trends closely resemble those reported in YbPtBi[41] and ZrTe$_5$[42], reinforcing the idea of a topological connection. We observe that the range of the magnetic field where the cubic-in-B Hall effect varies coincides with the range where the longitudinal magnetoresistance decreases quadratically before saturation (Fig. 4e). In the Supplementary Information, we show that the same semi-classical theory that leads to (1) predicts that the leading contributions to the cubic-in-$B$ Hall effect are due to the orbital magnetic moment and the Berry curvature of the multifold nodes. Since both diverge close to the multifold nodes, a non-linear cubic Hall is expected to be

sizable where the positive magnetoconductance is sizable, as seen in our experiment. Concretely, we find that the dominant contribution is the filled linear band at Γ, as its Fermi momentum is closest to the multifold crossing.

## Discussion

In summary, our measurements confirm the existence of negative longitudinal magnetoresistance in single-crystal CoSi, which we attribute to the chiral anomaly of multifold fermions. Our theory shows that the orbital magnetic moment contributions work against the chiral anomaly terms but do not overcome them, allowing negative magnetoresistance to be observable in our experiment. Additionally, the orbital and Berry curvature contributions dominate the cubic-in-magnetic field Hall term, acting as an additional signature of multifold fermions. While high-energy physics phenomena have been previously observed in analog condensed matter systems, our results mark the observation of a quantum anomaly of particles with no counterpart in high-energy physics, as multifold fermions are forbidden to exist as elementary particles. Our work showcases that quantum materials are a fruitful avenue to formulate and observe the most general physical phenomena linked to quantum anomalies.

## Methods

### Crystal growth

CoSi single crystals were grown in Te-flux. The starting materials Co (99.95%, 20 Alfa Aesar), Si (99.999%, Chempur) and Te (99.9999%, Alfa Aesar) were mixed in the molar 21 ratio of 1:1:20 and heated to 1050° C at a rate of 100° C/h and held there for 15 h. Successively, the sample was cooled to 700° C at a rate of 2° C/h, and extra Te-flux was removed by centrifugation. High-quality CoSi single crystals in the mm-range resulted from this growth protocol.

### Sample fabrication

From the CoSi bulk sample, a 2 $\mu$m × 1.5 $\mu$m × 15 $\mu$m lamella was cut and then patterned to a Hall bar shape with a focused ion beam system (FIB) of the type FEI Helios 600i using 30 keV Ga+ ions. The micro Hall bar was then transferred onto a patterned chip with Au contact pads (150 nm Au + 10 nm Ti, for adhesion) and welded using ion-assisted deposition of Pt.

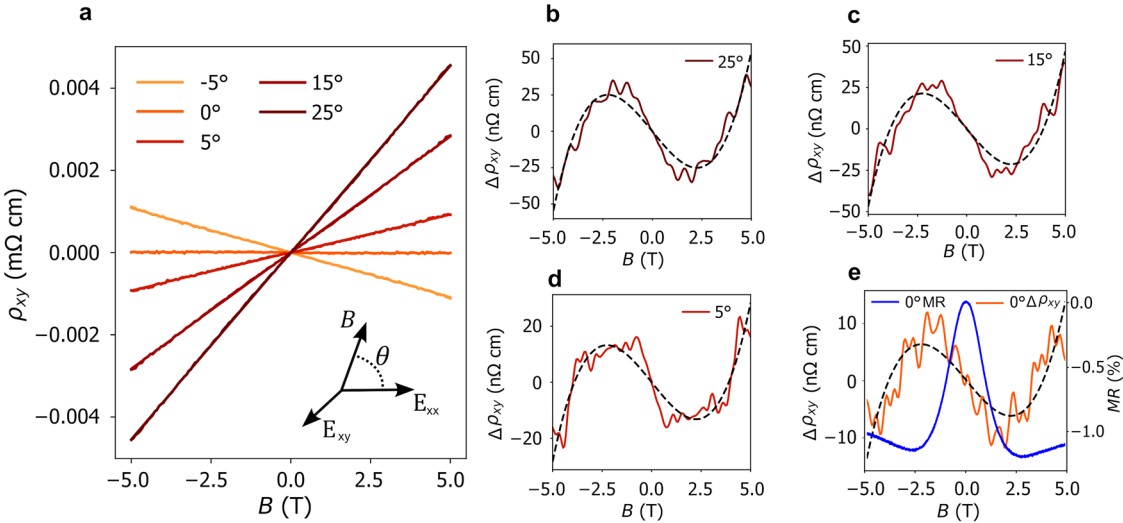

**Fig. 4 | Cubic-in-$B$ Hall effect. a** Angle-dependent Hall effect. **b–d** A non-linear-in-$B$ Hall effect is visible after subtracting the linear component from the Hall effect. The remaining contribution is well described by cubic polynomial - $\alpha B + \beta B^3$, as indicated by the dashed black lines. The negative longitudinal magnetoresistance is plotted on top (**e**) showing that the field range where the cubic component of the Hall effect is non-zero coincides with the range of decreasing negative magnetoresistance.

### Electrical transport measurements

Electrical measurements were performed in a cryostat (Dynacool from Quantum Design) using external lock-in amplifiers (MFLI from Zurich Instruments). The electrical current is always applied along the [110] direction. B oriented in [113] or [110].

### Scanning Transmission Electron Microscopy

The STEM measurements have been performed with a double spherical aberration corrected JEOL ARM200F microscope operated at 200 kV.

### Shubnikov–de Haas oscillations analysis

Shubnikov–de Haas oscillations have been isolated by subtracting a $4^{th}$ order polynomial fit of the magnetoresistive data, from 3.5–9 T. The power spectral density has been found by performing a fast Fourier transform on the oscillations plotted as $1/B$.

## Data availability

The data generated in this study have been deposited in the Zenodo database under accession code https://doi.org/10.5281/zenodo.12627039.

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

## Acknowledgements

A.G.G. is grateful to J. H. Bardarson, J. Behrends, and D. Pesin for insightful discussions. A.G.G., C.Z., A.M., and B.G. acknowledge financial support from the European Union Horizon 2020 research and innovation program under grant agreement No. 829044 (SCHINES). F.B. and B.G. acknowledge the SNSF project HYDRONICS under the Sinergia grant (No.189924). A.M. acknowledges funding support from the European Union's Horizon2020 research and innovation program under the Marie Sklodowska-Curie Grant Agreement No. 898113 (InNaTo). We are grateful to Philip Moll for sharing insights and support in FIB-based microstricturing. We wish to acknowledge the support of the Cleanroom Operations Team of the Binning and Rohrer Nanotechnology Center (BRNC). Continuous support from Ilaria Zardo, Heike Riel, Mark Ritter, and Kristin Schmidt is gratefully acknowledged.

## Author contributions

F.B., A.M., and B.G. conceive the experiment. A.G. developed the theory. C.F., V.H. grew the crystals. H.S. fabricated the sample. M.S. performed STEM imaging and analysis. F.B. performed the measurements and data analysis. F.B., L.R., C.Z., H.S., A.G., and B.G. interpreted the data. F.B. A.G. and B.G. wrote the manuscript with inputs from all authors.

## Competing interests

The authors declare no competing interests.
