## [Peer Review File · Nature Communications]

REVIEWER COMMENTS

Reviewer #1 (Remarks to the Author):

The authors report the observation of intrinsic longitudinal negative magnetoresistance in single crystalline CoSi featuring multifold bands within ~ 0.85 eV around the Fermi level, which is attributed to the chiral anomaly of multifold fermions. A semiclassical theory was proposed to interpret that the negative magnetoresistance originates in the chiral anomaly, although the orbital magnetic moment contributes contrary to the chiral anomaly. Additionally, the orbital and Berry curvature dominate the cubic-in-magnetic field Hall effect, acting as an additional signature of multifold fermions. Overall, I think it is a nice work of general interest and would recommend its publication on the Nature Communications after the authors address the following issues.

1. On page 5, Figure 1, please supply the experimentally observed band structure of CoSi single crystal by using the angle-resolved photoemission spectroscopy (like Nature 567, 496-499 2019 and so on) and compare it with the calculated one in Figure 1a.
2. On page 9, paragraph 1, “Finally, while an excess of Cobalt could contribute to the negative MR, because of its magnetic properties, such an occurrence would typically coincide with hysteresis in the MR or Hall effect data [47], which is not evident within our experimental error.” Here, “an excess of Cobalt” should be “an excess of Cobalt in CoSi (Co_{1+x}Si)”.

Reviewer #2 (Remarks to the Author):

It is a good paper and should be published after some corrections made and proper responses to the remarks.

1. Authors should finally decide how they grow the CoSi crystal. Two contradictory techniques mentioned in lines 93,94,103-105.
2. Exponent n in the dependence $\Delta\rho/\rho \sim H^n$ for CoSi as a rule differs from canonic value $n=2$ and is often equal to 1.8-1.9. Line 115. Authors should comment. See Ref. [36] and below:
D. Takane, Z.Wang, S. Souma, K. Nakayama, T. Nakamura, H.Oinuma, Y. Nakata, H. Iwasawa, C. Cacho, T. Kim, K. Horiba, H. Kumigashira, T. Takahashi, Y. Ando, and T. Sato,
Phys. Rev.Lett. 122, 076402 (2019).
S. Shanmukharao Samatham, D. Venkateshwarlu, and V.Ganesan, Mater. Res. Express 1, 026503 (2014).

X. Xu, X. Wang, T. A. Cochran, D. S. Sanchez, G. Chang, I. Belopolski, G. Wang, Y. Liu, H.-J. Tien, X. Gui, W. Xie, M. Z. Hasan, T.-R. Chang, and S. Jia,

Phys. Rev. B 100, 045104 (2019).

3. I cannot see any asymmetric longitudinal magnetoresistance in Ref.[36]. Line 129-132. Yet, in Fig.4 one can see somewhat strange behavior of the longitudinal MR of the high resistivity sample of CoSi, probably indicating a tendency to negative value. Another sample used in Ref.[36] with resistivity $\rho = 14.9 \mu\text{Ohm cm}$ at 2 K, and $\text{RRR}=9.6$ do not show that kind of tendency. It has a size of approximately $0.4 \times 1.1 \times 9.5 \text{ mm}^3$.

Reference [34] doesn't contain any information on longitudinal MR in CoSi.

Ref. [37] reports a negative longitudinal MR in CoSi doped with Fe, but not in pure CoSi.

See also: Lauritz Schnatmann, Michaela Lammel, Christine Damm, Aleksandr A. Levin, Nicolas Pérez, Sergey Novikov, Alexander Burkov, Heiko Reith, Kornelius Nielsch, Gabi Schierning, *Advanced Electronic Materials* 8, 2101081 (2022).

All of these need discussion and comments.

Reviewer #3 (Remarks to the Author):

This manuscript reports measurements of negative longitudinal magnetoresistance in CoSi, which the authors attribute to the chiral anomaly of multifold chiral fermions, present in its bandstructure.

Current jetting is claimed to be ruled out by the "squeezing test", proposed earlier by N.P. Ong. The paper is generally well-written and clear and the results seem reasonable, although not

really unexpected and not particularly interesting in my view. Nevertheless, I still think this is a valuable paper overall and should be published. One thing I would suggest the authors discuss more

clearly is the significant difference in magnitude of the effect between CoSi in this paper and earlier measurements by N.P. Ong in Na₃Bi. In the latter case, the effect was large, several hundred percent.

In contrast, the magnitude reported here is 1%, even though the chiral fermions are well-separated in the Brillouin zone. What is the origin of such a large difference in magnitude?

Dears,

We are thankful for the valuable reports and the constructive remarks. We attach the new version of the manuscript with the changes highlighted in blue. Below, you find our point-by-point responses to your concerns.

Sincerely,

The authors

REVIEWER 1

The authors report the observation of intrinsic longitudinal negative magnetoresistance in single crystalline CoSi featuring multifold bands within 0.85 eV around the Fermi level, which is attributed to the chiral anomaly of multifold fermions. A semiclassical theory was proposed to interpret that the negative magnetoresistance originates in the chiral anomaly, although the orbital magnetic moment contributes contrary to the chiral anomaly. Additionally, the orbital and Berry curvature dominate the cubic-in-magnetic field Hall effect, acting as an additional signature of multifold fermions. Overall, I think it is a nice work of general interest and would recommend its publication on the Nature Communications after the authors address the following issues.

- 1. On page 5, Figure 1, please supply the experimentally observed band structure of CoSi single crystal by using the angle-resolved photoemission spectroscopy (like Nature 567, 496-499 2019 and so on) and compare it with the calculated one in Figure 1a.*

Reply: We thank the referee for the comments. We agree on the importance of the experimental confirmation of the calculated band structure. Therefore, we have made sure that our calculation are consistent with the ones that have already been published and confirmed by ARPES measurements, see for example Fig 1 in D. Takane, et al, Phys. Rev. Lett. 122, 076402. As their study focuses on ARPES in CoSi, we adopted their findings and

concentrated on the transport study of this material. Still, to make sure that our sample complies to such band structure we have compared the measured quantum oscillation's frequencies with the ones calculated for CoSi, demonstrating overall consistency.

Changes: We have changed the text to emphasize the consistency with ARPES, highlighted in page 4 line 91:

This band-structure has been confirmed by ARPES measurements [17, 19] and...

2. On page 9, paragraph 1, “Finally, while an excess of Cobalt could contribute to the negative MR, because of its magnetic properties, such an occurrence would typically coincide with hysteresis in the MR or Hall effect data [47], which is not evident within our experimental error.” Here, “an excess of Cobalt” should be “an excess of Cobalt in CoSi ($Co_{1+x}Si$)”.

Changes: We thank the referee for this precision. We have changed the text as suggested, highlighted in page 9, lines 159-162:

Finally, while an excess of Cobalt in CoSi ($Co_{1-x}Si_x$) could contribute to the negative MR, because of its magnetic properties, such an occurrence would typically coincide with hysteresis in the MR or Hall effect data.

REVIEWER 2

It is a good paper and should be published after some corrections made and proper responses to the remarks.

1. Authors should finally decide how they grow the CoSi crystal. Two contradictory techniques mentioned in lines 93,94,103-105.

Reply: We thank the referee for carefully reading the manuscript and for the useful comments.

For our experiment we use chemical vapor transport method as mentioned in lines 93,95. The reference reported in line 103,105 ([27] X. Xu, et al Phys. Rev. B 100, 045104 (2019)) shows measurements of several CoSi sample, grown using different techniques and conditions. Our sample should be compared with sample I04 of ref [27], which has the same transport direction (110), same growth method (CVT), similar growth temperature (1100° vs ours

1050°). They report $RRR = 7.33$ and $MR(9T) = 29\%$ for bulk sample. We have measured $RRR = 6$, $MR = 25\%$, in our microstructured sample.

Changes: We have changed the text to avoid any ambiguity, page 4 lines 104-106:

which aligns with the literature values from bulk crystals, with similar growth conditions and same transport direction (see for example sample I04 in [27]).

2. *Exponent n in the dependence $\Delta\rho/\rho_0 \approx H^n$ for CoSi as a rule differs from canonic value $n=2$ and is often equal to 1.8-1.9. Line 115. Authors should comment. See Ref. [36] and below: D. Takane, Z.Wang, S. Souma, K. Nakayama, T. Nakamura, H.Oinuma, Y. Nakata, H. Iwasawa, C. Cacho, T. Kim, K. Horiba, H. Kumigashira, T. Takahashi, Y. Ando, and T. Sato, *Phys. Rev.Lett.* 122, 076402 (2019). S. Shanmukharao Samatham, D. Venkateshwarlu, and V.Ganesan, *Mater. Res. Express* 1, 026503 (2014). X. Xu, X. Wang, T. A. Cochran, D. S. Sanchez, G. Chang, I. Belopolski, G. Wang, Y. Liu, H.-J. Tien, X. Gui, W. Xie, M. Z. Hasan, T.-R. Chang, and S. Jia, *Phys. Rev. B* 100, 045104 (2019).*

Reply: The exponent n in the dependence $\Delta\rho/\rho_0 \approx B^n$ for CoSi usually slightly deviates from $n = 2$. In our sample a fit of the MR as $MR = \alpha B^n$ ($\alpha = 0.34 \text{ Tesla}^{-n}$ and $n = 1.94$) fits the data slightly better than $MR = \alpha B^2$ ($\alpha = 0.31 \text{ Tesla}^{-2}$), still having an extra free parameter (see Figure 1 in this reply). For the scope of this work, we concentrated on the smoothness of the MR, rather than the exact exponent. While we do not know why the field exponent varies from 1.8 to 2 in different reports, it is not unreasonable to speculate that the scattering time can be weakly magnetic-field dependent, changing the canonical picture. Since the deviation from $n = 2$ is small, we believe this does not invalidate our conclusions.

3. *I cannot see any asymmetric longitudinal magnetoresistance in Ref.[36]. Line 129-132. Yet, in Fig.4 one can see somewhat strange behavior of the longitudinal MR of the high resistivity sample of CoSi, probably indicating a tendency to negative value. Another sample used in Ref.[36] with resistivity $\rho = 14.9 \mu\text{Ohm cm}$ at 2 K, and $RRR=9.6$ do not show that kind of tendency. It has a size of approximately $0.4 \times 1.1 \times 9.5 \text{ mm}^3$. Reference [34] doesn't contain any information on longitudinal MR in CoSi. Ref. [37] reports a negative longitudinal MR*

Figure 1. Fit of the trasverse MR using $MR \approx B^2$ and $MR \approx B^n$. From the fit on the right $n = 1.94$.

in CoSi doped with Fe, but not in pure CoSi. See also: Lauritz Schmatmann, Michaela Lammel, Christine Damm, Aleksandr A. Levin, Nicolas Perez, Sergey Novikov, Alexander Burkov, Heiko Reith, Kornelius Nielsch, Gabi Schierning, Advanced Electronic Materials 8, 2101081 (2022). All of these need discussion and comments.

Reply: In Ref [36] the asymmetry in the MR has been corrected. At the end of the Experiment paragraph: *The results of the magnetoresistance measurements, corrected for the Hall contributions arising due to the sample misalignments and not-quite-correct positions of the electrical contacts, are illustrated in Figs. 3 and 4.* The raw data for sample Cos-1 are shown in the ArXiv version (Fig. 3 in arXiv:2209.02036), the raw data for sample Cos-2, to our knowledge, have not been published. We believe that the correct position of the electrical contacts is crucial for proper measurements. Any misalignment would add the contribution from the Hall signal or the transversal MR signal, which easily overcome the small negative MR effect. In our experiments we see a negative magnetoresistance in the raw data, without the need of symmetrization or filtering. Also, we employ FIB microfabrication specifically to have "good control of geometry and crystalline direction", and avoid spurious effects in our longitudinal magnetotrasport experiments.

The negative LMR observed in reference [34] (shown in Figure 3c) is difficult to include in a more detailed discussion, because no claim is made and other possible origins of negative

LMR are not systematically excluded. This agrees with private communication with the authors.

Finally, we report that a negative MR has been measured in Fe-doped CoSi ref [37]. This work is noteworthy because the authors claim that their measurements indicate the presence of a chiral anomaly in Fe-doped CoSi. However, there are no predictions for the chiral anomaly in Fe-doped CoSi. Moreover, to fit the MR data, the authors used a model that involves a combination of chiral anomaly and weak anti-localization, demonstrating that their sample is not "clean". Consequently, we believe that the findings of Schnatmann et al. are insufficient to conclusively demonstrate the chiral anomaly in multifold fermions. Therefore, we have sought to advance this area of research with our own proposed study.

Changes: We have changed the text to be more neutral, page 8 lines 128-133:

It is worth noting that previous research reported both positive longitudinal MR in CoSi single crystal [36] (yet asymmetric with the magnetic field), and negative longitudinal MR (in CoSi single crystal in [34] and in Fe-doped CoSi in [37].) It appears for a convincing claim on the observation of the anomaly, more detailed cross-checks and modeling is necessary, which we aim to provide in the following.

REVIEWER 3

This manuscript reports measurements of negative longitudinal magnetoresistance in CoSi, which the authors attribute to the chiral anomaly of multifold chiral fermions, present in its bandstructure. Current jetting is claimed to be ruled out by the "squeezing test", proposed earlier by N.P. Ong. The paper is generally well-written and clear and the results seem reasonable, although not really unexpected and not particularly interesting in my view. Nevertheless, I still think this is a valuable paper overall and should be published. One thing I would suggest the authors discuss more clearly is the significant difference in magnitude of the effect between CoSi in this paper and earlier measurements by N.P. Ong in Na3Bi. In the latter case, the effect was large, several hundred percent. In contrast, the magnitude reported here is 1%, even though the chiral fermions are well-separated in the Brillouin zone. What is the origin of such a large difference

in magnitude?

Reply: We thank the referee for the comment. Indeed, given the large separation of the chiral branches one would expect large effects related to the chiral anomaly. This is in part true. In fact, according to our analysis, we find a "chiral scattering time" that exceed the Drude scattering time by a factor of 150. This ratio is 3 to 4 times larger than what N.P. Ong et al. estimated in Na₃Bi. This is consistent with the large node separation.

However, it turns out that the the main contribution for the negative magnetoresistance induced by the chiral anomaly in CoSi comes from the tiny pocket at Gamma. This is because the Fermi level is very close to the node at Gamma, making the Berry curvature contribution sizable compared to that of the R point, which acts like a large metallic pocket. This justifies the small magnitude (1%) measured. Indeed, according to our calculation (see Equation 1, page 9), the long scattering time τ_v , is compensated by the small ratio between the Fermi momenta: $r_1 = \left(\frac{k_F^\Gamma}{k_F^R}\right)^2 \frac{v_R}{v_\Gamma}$, leading overall to a small negative magnetoresistance.

Changes: We have changed the text to explicitly compare our results with the original results on Na₃Bi, highlighted in page 10, lines 204-213:

For comparison, in their original work on the chiral anomaly in Na₃Bi, Xiong et al. found a axial current relaxation time that exceeds the Drude relaxation time by a factor of 40-60, yet leading to a larger positive magnetocundactance when compared to CoSi (25% in Na₃Bi against the 1% of CoSi). This apparent discrepancy arises because the main contribution to the positive magnetocundactance induced by the chiral anomaly in CoSi comes from the tiny pocket at Γ . The Fermi level is very close to the node at Γ , making the Berry curvature contribution significant compared to that of the R point, which behaves like a large metallic pocket. Indeed, according to Eq. 1 the long scattering time τ_v , is compensated by the small ratio $r_1 = \left(\frac{k_F^\Gamma}{k_F^R}\right)^2 \frac{v_R}{v_\Gamma}$, leading overall to a small positive magnetoconductance.

REVIEWERS' COMMENTS

Reviewer #1 (Remarks to the Author):

The authors have addressed the referee's concerns. Now the manuscript could be published.

Reviewer #2 (Remarks to the Author):

I believe that the authors have addressed all my remarks and the manuscripts should be published.

Reviewer #3 (Remarks to the Author):

The authors have responded to my comments in a satisfactory way and the paper may now be published.